# Effect of Waterlogging Stress on Leaf Anatomical Structure and Ultrastructure of *Phoebe shareri* Seedlings

Fenghou Shi [1,*], Zhujing Pan [1], Pengfei Dai [1], Yongbao Shen [1], Yizeng Lu [2] and Biao Han [2]

1   Collaborative Innovation Centre of Sustainable Forestry in Southern China, College of Forestry,
    Nanjing Forestry University, Nanjing 210037, China; panzj@njfu.edu.cn (Z.P.); xd007@njfu.edu.cn (P.D.);
    ybshen@njfu.edu.cn (Y.S.)
2   Shandong Provincial Center of Forest and Grass Germplasm Resources Resources, Jinan 250102, China;
    luyizeng@shandong.cn (Y.L.); hanbiaok831228@163.com (B.H.)
*   Correspondence: shifenghou@njfu.edu.cn

**Abstract:** *Phoebe shareri* is an excellent roadside tree with a wide distribution range and high ornamental value. Excessive moisture can affect the external morphology, the microstructure, and the ultrastructure of the leaf. Little is known at present regarding the leaf structure of *P. shareri* under waterlogging stress. In this paper, the external morphology of leaves, the microstructure of leaf epidermis, and the ultrastructure of mesophyll cells of *P. shareri* seedlings under waterlogging stress and drainage were dynamically observed. Waterlogging stress contributed to the yellowing and wilting of *P. shareri* seedling leaves, the gradual closing of leaf epidermal stomata, increasing density of leaf stomata, gradual loosening of the arrangement of leaf cell structure, and merging of leaf palisade tissue cells. Waterlogging stress forced the structure of the chloroplast membranes to blur, gradually causing swelling, and deformation, with plasmolysis occurring in severe cases. During waterlogging, the basal lamellae were disorganized, and the mitochondrial membrane structure was damaged. The damaged state of the leaves was not relieved after drainage. Waterlogging stress not only inhibited the growth of leaves, but also accelerated the closure of stomata, disordered the arrangement of palisade tissue and spongy tissue gradually, and damaged the internal organelles of mesophyll cells.

**Keywords:** *Phoebe shareri*; waterlogging stress; leaf anatomical structure; leaf ultrastructure

## 1. Introduction

Climate change affects atmospheric precipitation, leading to extreme weather events and an increase in the frequency and duration of floods [1,2]. During the plum rain season in the middle and lower reaches of the Yangtze River, China, there are extended periods of continuous heavy rainfall, which severely hinder seedling growth [3]. Moisture is one of the essential elements in the growth of seedlings, but waterlogging stress can arise from saturation or even excessive water in the soil [4]. Excessive water directly affects the diffusion of oxygen in plant tissue, forming an oxygen-deficient environment, and heavily impairing gas exchange between plants and the atmosphere [5]. Prolonged waterlogging and hypoxic environments ultimately lead to the accumulation of toxic metabolites and the increase of reactive oxygen species, affecting the physiological and biochemical processes of seedlings, and hindering the growth of seedlings, and, thus, eventually leading to cell death and plant senescence [6–8]. Although most plants perform poorly when waterlogged, they can resist waterlogging stress by changing their morphological structures, physiological characteristics, cell structures, etc. [5,9].

Under waterlogging conditions, the appearance of leaves may show signs of wilting, curling, or even abscission [10]. When 15~25-cm-tall *Celosia argentea* and *Melampodium paludosum* were waterlogged for ten days, the leaf of *M. paludosuma* appeared to droop

after three days, and those of *C. argentea* appeared to droop after 7 days, and both had no ornamental value after the waterlogging treatment ended [11].

The effects of waterlogging stress on seedlings are not only reflected in the changes in external morphology, but also in changes in the internal cell structure of seedling leaves, such as the destruction of cell membrane structure, swelling and deformation of chloroplasts, loose arrangement of grana lamellae, and extravasation of mitochondrial contents [12–14]. Seedlings can cope with waterlogging stress by controlling stomatal closure level and leaf gas exchange rate, which can be reflected in the dynamic changes of leaf anatomical structure [15]. In the previous study, after seedlings of *Sorghum bicolor* were waterlogged for two weeks, leaf thickness significantly decreased, mainly due to changes in the thickness of the upper epidermal and mesophyll cells [16]. When *Avicennia marina* was waterlogged by means of experimentally simulated semidiurnal tides, it was observed that, with increasing duration of waterlogging, leaf thickness, mesophyll thickness, palisade parenchyma thickness, palisade–spongy ratio, and hypodermis thickness decreased, but the mesophyll to leaf thickness ratio increased [17]. After deep waterlogging treatment (with a waterlogging depth exceeding 20 cm of the soil layer), the leaf structure of *Quercus shumardii* blurred, some palisade tissue cells became irregular, becoming almost round or oval, the gap of sponge tissue cells became larger, and some lower epidermal cells disintegrated [18].

In addition, the morphology and structure of organelles. such as chloroplasts and mitochondria, in mesophyll cells change under waterlogging stress, which plays an important role in leaf photosynthesis [4]. In the study, a waterlogged pool of *Zea mays* was maintained at 2–3 cm above the soil surface. During the different waterlogging stages, the cytoderm structure of *Z. mays* became incomplete and exhibited indistinct gradation, irregular chloroplast configuration, reduction in chloroplast numbers in mesophyll cells, and dimming of the mitochondrial outer membrane. Fine-grained substance exosmosis occurred and structures became diluted after exposure to waterlogging [12]. Under waterlogging treatment, there were almost no substances in the cells of *Chionanthus virginicus* leaves, due to the chloroplasts degrading, starch particles accumulating in large quantities, and severe damage to the structure of the cells [19]. The chloroplasts of *Carya illnoinensis* showed similar changes after waterlogging for 30 days, in that the chloroplasts were swollen and rounded, the number of osmophilic particles increased, and the lamella structure began to loosen [20]. Thus, it is important to investigate the effects of waterlogging on leaves in terms of external morphology, anatomical structure and ultrastructure.

As an evergreen *Phoebe* tree in the Lauraceae family, *Phoebe sheareri* is suitable for places with warm, humid climates and fertile soil. *P. sheareri* has a long growth cycle, a tall and beautiful tree shape, and dense branches and leaves, and is an excellent shade tree and roadside tree with high ornamental value [21,22]. In addition, *P. sheareri* wood is hard, straight textured, corrosion-resistant, and durable, and has a special fragrance. It is is an excellent material for high-end furniture, architecture, and wooden utensils, and is the royal building material of choice for palaces, and tombs and for their gardens [23]. The phenomenon of waterlogging often occurs in the Yangtze River Delta region, which is an area where *P. sheareri* is widely distributed, so it is important to study the response of *P. sheareri* seedling leaves to waterlogging [24]. Currently, studies on the stress resistance of *P. sheareri* have mainly focused on low-temperature stress [25] and drought stress [26], and less research has been connducted on waterlogging stress.

Few studies have reported on the effects of different durations of waterlogging and drainage on the leaf morphology and structure of *P. sheareri*. In this study, we used the "double pot method" to artificially simulate the waterlogging experimental environment. The aims were the following: (1) to measure changes in the leaf morphology, leaf epidermal stomata, leaf anatomical structure and ultrastructure of *P. sheareri* under waterlogging and drainage; (2) to understand the changes and coping mechanisms of the leaf structure of *P. sheareri* seedlings in a waterlogging environment; (3) to provide basic waterlogging informational support for the promotion and application of the *P. sheareri* species.

Based on the results of the above-related studies, before the experiment began, we speculated that, during waterlogging treatment, the leaf morphology of *Phoebe sheareri* seedlings would exhibit gradual yellowing, falling off, and even dying. The leaf anatomical structure would gradually change, with the width and opening and closing degrees of the leaf epidermal layer becoming smaller, the palisade tissue cells becoming irregular and nearly round or oval, the cell spaces becoming larger, and the leaf thickness decreasing. In leaf ultrastructure, the chloroplast and mitochondrial organelles would be damaged, the chloroplast shape would change, gradually swelling and becoming round, the lamellar structure would gradually loosen, the mitochondrial membrane would be damaged, and the contents would exude. We further speculated that the damaged states of leaf morphology, anatomical structure and ultrastructure might not improve after drainage treatment of *Phoebe sheareri* seedlings after the waterlogging.

## 2. Materials and Methods

### 2.1. Plant Materials

The experimental materials were 2-year-old container seedlings of *P. sheareri*. The seeds were collected from the wild population of Qionglong Mountain, Suzhou, Jiangsu Province. In the middle of June 2020, container seedlings, with basically the same growth, were selected for pot change transplanting. The transplanting container was a plastic flowerpot with a height of 20 cm, an upper diameter of 23 cm, and a lower diameter of 19 cm, The seedling substrate was produced by Jiangsu Xingnong Substrate Technology Co., Ltd., (Zhenjiang, China) with a dry bulk density of 0.4 g/cm$^3$, total porosity of 64%, large to small porosity ratio of 1:5, pH 7.6, EC 3.2, organic matter content of 41%, and total nitrogen, phosphorus, and potassium content of 3.8%. When transplanting, the seedling substrate from the original containers was gently washed away to preserve the entire root system of the seedlings as much as possible; An equal amount of seedling substrate was put into each plastic flowerpot, and one seedling transplanted to each container. Sufficient root fixing water was poured in. After transplanting, the seedlings were placed in the greenhouse of Baima Teaching and Scientific Research Base of Nanjing Forestry University (Baima Town, Lishui District, Nanjing, China) for slow seedling growth, during which normal water and fertilizer management was carried out.

### 2.2. Waterlogging Treatment

To explore the leaf morphology, anatomical structure and ultrastructure changes in *P. sheareri* under waterlogged and waterlogged–drainage conditions, we used the "double pot method" to artificially simulate the waterlogging environment.

This experiment adopted a single factor completely randomized block design and consisted of two parts: a waterlogging experiment and a waterlogging–drainage experiment. A total of 180 plants, with roughly the same growth potential, were used in the two-part experiment, from which 30 seedlings were randomly selected for each treatment, and three biological replicates were applied. A total of 120 seedlings were divided into four waterlogging experiments. A total of 60 seedlings participated in the drainage process, being waterlogged for 12 days and drained for 6 days and 12 days, respectively. The seedlings under normal management and without waterlogging treatment were used as the control group (Table 1).

**Table 1.** Each treatment of *P. sheareri* seedlings for waterlogging and waterlogging-drainage experiments.

| Experiment | Treatment |
|---|---|
| Part 1 Waterlogging treatment | Control group (without waterlogging treatment)<br>Waterlogging stress for 6 days<br>Waterlogging stress for 12 days<br>Waterlogging stress for 18 days |
| Part 2 Waterlogging-drainage treatment | Control group (without waterlogging treatment)<br>Waterlogging stress for 12 days<br>Waterlogging stress for 12 days and drainage for 6 days<br>Waterlogging stress for 12 days and drainage for 12 days |

Three biological replicates were applied for each treatment.

The waterlogging treatment began on 11 September 2020, named the 0-day for waterlogging stress. In adopting the "double pot method", the seedlings of *P. sheareri* were placed in a lotus basin together with a plastic pot, and tap water was poured in to obtain a water level approximately 2–5 cm above the substrate (Figure 1). Throughout the test, the water level change was observed, and timely replenishment maintained the submergence level. The waterlogging-drainage treatment involved removing seedlings treated with waterlogging for 12 days from their lotus pots and draining the substrate naturally without water.

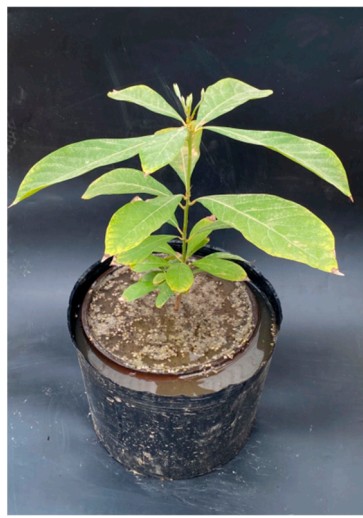

**Figure 1.** Schematic diagram of "double pot method".

*2.3. Indicator Determination*

Plant sampling: we observed and photographed the overall shape and morphological changes of seedlings during the treatment of waterlogging stress and drainage. At 9:00 a.m. on the sampling day, 3–5 standard plants were selected, and the 3rd to 5th intact functional leaves from the bottom to the top of the seedlings were collected, and washed with tap water several times to remove any dirt and dust accumulated on their surfaces. Small circular pieces, about 0.5 cm wide, from the central part of the leaves (avoiding the main veins), were taken as samples, put into glutaraldehyde fixative solution, and stored at 4 °C, so as to observe the changes in the stomata, anatomical structure and microstructure of the seedling leaves.

Seedling leaf morphological observation: we observed the morphological changes of *P. sheareri* seedlings and their leaves, and took pictures to record the growth status of seedlings and leaves under different waterlogging and drainage periods.

Stomata and anatomical structure of seedling leaf epidermis: referring to the method of Jain et al. [27], the leaf stomata and anatomical structure of *P. sheareri* seedlings were observed, recorded and photographed using Scanning Electron Microscopy (SEM).

The samples were taken out of the glutaraldehyde, and rinsed to remove traces of glutaraldehyde. The fixed samples were dehydrated through a series of dehydrating solutions (gradually dehydrated at 30%, 70%, 80%, and 90% ethanol gradients), and subjected to critical point drying. The dehydrated samples were fixed on an adhesive platform and sputtered with a film of gold using an ion-beam sputter coater. The samples were examined under an FEI Quanta 200 Scanning Electron Microscope (USA) at Nanjing forestry university, Nanjing, China. The stomatal length, stomatal width, stomatal opening rate, and stomatal number per unit area were observed and photographed under a 600× microscope, and were measured by Image J (a free, Java-based image-processing package, version 1.8.0.112). This process was repeated 3 times, and 5 visual fields were randomly observed to calculate the average value.

Microstructure of seedling mesophyll cells: referring to the method of Salah et al. [28], the internal ultrastructure of mesophyll cells of *P. sheareri* was observed, recorded and photographed with a transmission electron microscope.

The samples were taken out and washed 3 times with 100 mmol/L phosphoric acid buffer at pH 7.4. They were fixed in 1% buffered osmium tetroxide for 1 h (this process was carried out in a fume hood) and then cleaned 3 times for 10 min each time. Then, the samples were placed in ethanol of different concentration gradients (30%, 45%, 60%, 75%) for dehydration treatment. Finally, the samples were treated with anhydrous ethanol and pure acetone for 30 min. After the above dehydration treatment, the samples were soaked in Spurr's epoxy resin (Sigma-Aldrich, St. Louis, MO, USA) for 12 h, and kept in a dark environment. Using an ultra-thin slicer, the samples were sliced into 50 nm sections. After staining, the samples were observed and photographed under transmission electron microscopy (JEM 2100 High-Resolution Transmission Electron Microscopy, Tokyo, Japan). This was repeated 3 times, and 3 visual fields randomly observed.

### 2.4. Statistical Analysis

A Shapiro–Wilk test, one-way analysis of variance (ANOVA) data analyses and Duncan multiple comparisons were performed with SPSS version 22.0, which were used to assess data normality and to analyze the differences in the waterlogging and waterlogging–drainage treatments, respectively. Comparisons with $p$ values less than 0.05 were considered significantly different (expressed in lowercase letters), and those with less than 0.01 were considered extremely significantly different (expressed in uppercase letters). The correlation analyses were performed with the Pearson model two-tail test.

## 3. Results

### 3.1. Changes in Leaf Morphology of Phoebe sheareri Seedlings during Waterlogging Stress and Drainage

With the extension of waterlogging stress time, the leaf morphology of *P. sheareri* seedlings changed (Table 2). New leaves sprouted on the seedlings of *P. sheareri* after 6 days of waterlogging stress. Under waterlogging stress for 12 days, some leaves drooped for a short time at noon or dusk, but the droop disappeared at night and early morning. However, after 18 days of waterlogging, the leaves wilted continuously and did not recover, which indicated that the leaves were severely damaged. The seedlings that had been waterlogged for 12 days were drained. When they were drained for 6 days, the leaves drooped, wilted, and became chlorotic. Nearly 50% of the seedlings died. When the seedlings were drained for 12 days, 50% of the leaf lost green coloring and turned yellow, 50% of the leaf withered and fell off, 83% of the *P. sheareri* seedlings died, the leaves did not recover to their states before waterlogging and showed more serious damage.

**Table 2.** Changes in leaf morphology of *Phoebe sheareri* seedlings under waterlogging stress.

| Experimental Treatment | Leaf Morphology |
| --- | --- |
| Control group | The seedlings grew normally with green leaves and no wilting and drooping phenomenon; |
| Waterlogging stress for 6 days | The seedlings and their leaves grew well with 2–3 new leaves on the top; |
| Waterlogging stress for 12 days | About 30% of the leaves appeared with yellow spots, and 50% of the leaves appeared with a temporary wilting phenomenon; |
| Waterlogging stress for 18 days | All leaves of the seedling wilted and drooped, and all of the terminal bud and basal leaves of the seedlings turned yellow and fell off; |
| Waterlogging stress for 12 days and drainage for 6 days | Nearly 50% of the leaves drooped, and 50% of the leaves lost green coloring and turned yellow |
| Waterlogging stress for 12 days and drainage for 12 days | All leaves withered and drooped, 50% of the leaves lost green coloring and turned yellow, and 50% of the leaves withered and fell off. |

*3.2. Changes in Leaf Epidermal Stomata of Phoebe sheareri Seedlings during Waterlogging Stress and Drainage*

3.2.1. Effects of Waterlogging Stress on Leaf Epidermis Stomata of *Phoebe sheareri* Seedlings

The stomata of *P. sheareri* seedling leaves are concentrated in the lower epidermis of the leaves, are mostly oval or nearly round, and the stomata apparatus is composed of two kidney-shaped guard cells (Figure 2a). The observation of stomata in the leaf epidermis of *P. sheareri* under waterlogging stress showed that, with the extension of stress time, stomata in the leaf epidermis gradually closed, and its opening degree gradually decreased (Figure 2).

With the extension of waterlogging stress time, the stomatal length in the leaves of *P. sheareri* seedlings did not change significantly, and the stomatal width showed a gradually decreasing trend (Tables 3 and A1). The stomatal width of seedlings under waterlogging stress for 6 days had extremely significant differences from those of other treatments ($p < 0.01$). Compared to waterlogging stress for 12 days and 18 days, the stomatal width increased by 2.56 μm and 2.85 μm, respectively, during waterlogging stress for 6 days. The effect of waterlogging stress on leaf stomatal opening rate reached an extremely significant level ($p < 0.01$) and gradually decreased with the extension of waterlogging time. Compared with the control group, the stomatal opening rate decreased by 14.52% after 18 days of waterlogging treatment. At 18 days of waterlogging stress, the stomatal width and stomatal opening rate of *P. sheareri* seedlings were the lowest, at 2.67 μm and 13.03%, respectively. Under the condition of waterlogging stress, the stomatal density of *P. sheareri* seedlings increased gradually, and there were extremely significant differences among different treatments ($p < 0.01$). The order, from large to small, was the following: waterlogging stress for 18 days > waterlogging stress for 12 days > waterlogging stress for 6 days > Control group).

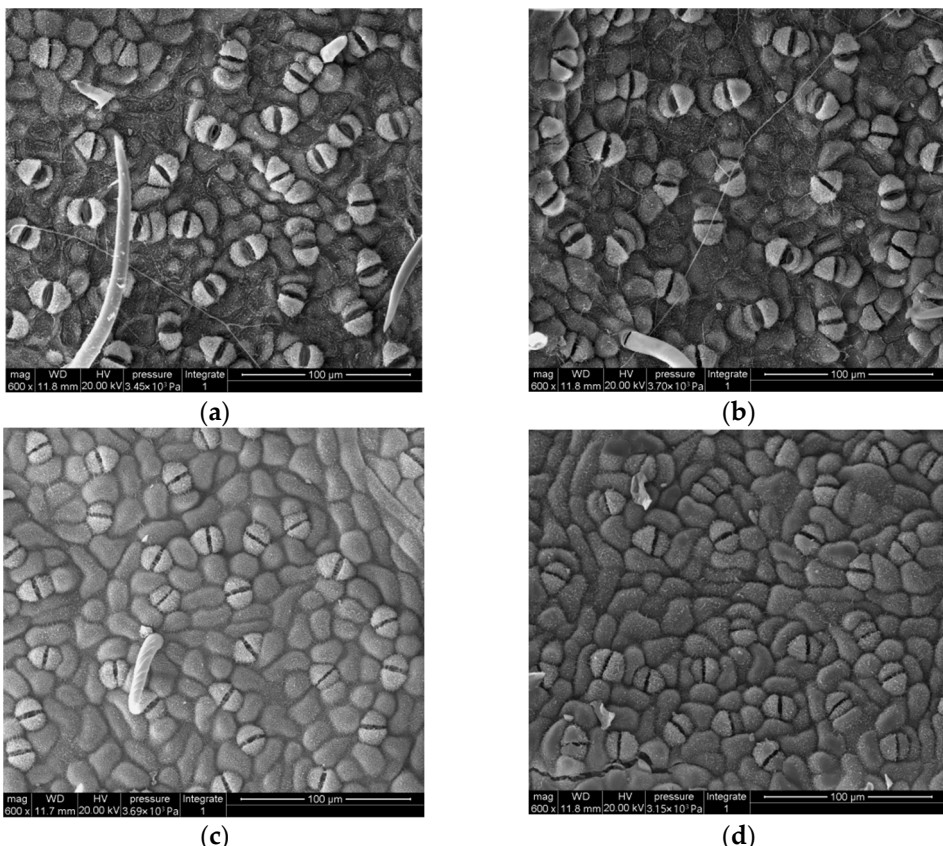

**Figure 2.** Changes in leaf epidermal stomata of seedlings under waterlogging stress. (**a**): Control group; (**b**): Waterlogging stress for 6 days; (**c**): Waterlogging stress for 12 days; (**d**): Waterlogging stress for 18 days.

**Table 3.** Determination results of leaf epidermal stomata of seedlings under waterlogging stress.

| Treatment | Stomatal Length (μm) | Stomatal Width (μm) | Stomatal Opening Rate (%) | Stomatal Density (n/mm²) |
|---|---|---|---|---|
| Control group | 13.35 ± 0.49 Aa | 7.35 ± 0.34 Aa | 27.55 ± 0.34 Aa | 2082 ± 22.47 Dd |
| Waterlogging stress for 6 days | 14.18 ± 0.38 Aa | 5.52 ± 0.28 Bb | 22.91 ± 0.06 Bb | 2336 ± 12.96 Cc |
| Waterlogging stress for 12 days | 14.58 ± 0.33 Aa | 2.96 ± 0.20 Cc | 17.41 ± 0.76 Cc | 3298 ± 75.58 Bb |
| Waterlogging stress for 18 days | 14.71 ± 0.66 Aa | 2.67 ± 0.11 Cc | 13.03 ± 0.25 Dd | 3467 ± 5.59 Aa |

Note: Different capital letters indicated extremely significant differences between treatments ($p < 0.01$), and different lowercase letters indicated significant differences between treatments ($p < 0.05$). The same applies to the below.

### 3.2.2. Effect of the Waterlogging-Drainage Process on Leaf Epidermal Stomata of *Phoebe sheareri* Seedlings

With the extension of drainage time, the stomatal opening degree and the number of stomata per unit area witnessed gradual decrease (Figure 3c,d). The leaf stomatal length of *P. sheareri* seedlings showed gradual increase after drainage (Tables 4 and A2), among which, the leaf stomatal length of the seedlings drained for 12 days had extremely significant differences from that of the control seedlings ($p < 0.01$), and had significant differences from that of the seedlings waterlogged for 12 days and drained for 6 days ($p < 0.05$). After waterlogging stress for 12 days and drainage for 12 days, leaf stomatal length reached a maximum (16.16 μm), with an increase of 2.81 μm compared with the control group.

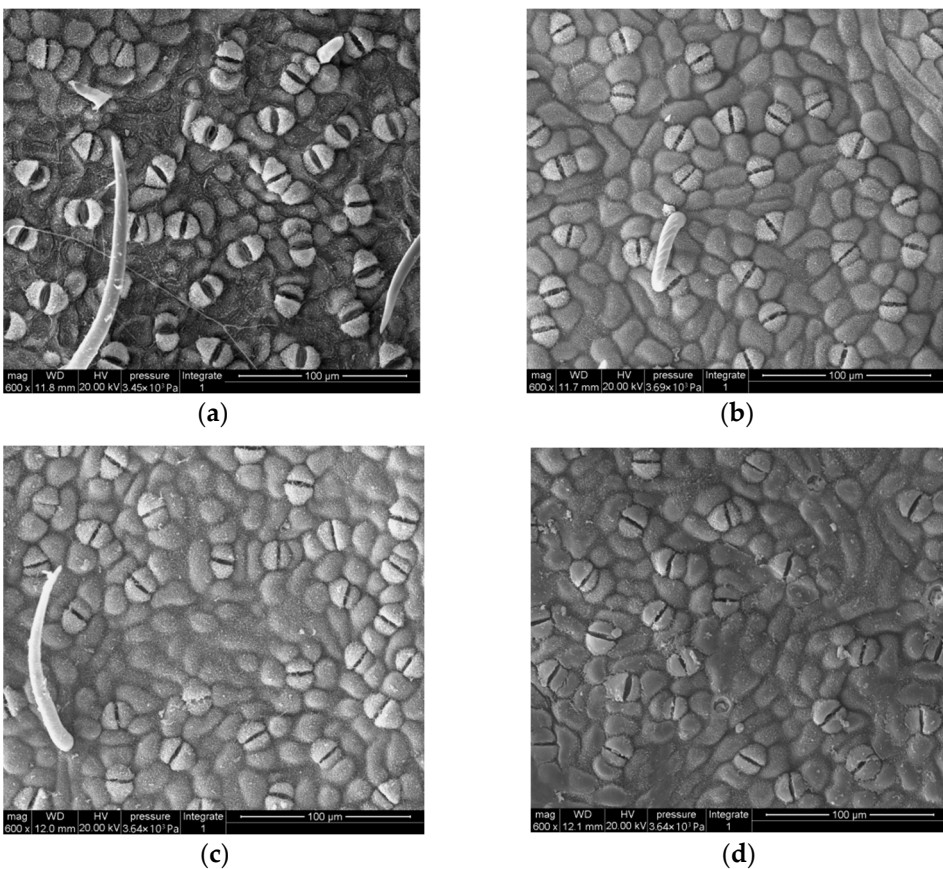

**Figure 3.** Changes in leaf epidermal stomata of seedlings during the waterlogging-drainage process. (**a**): Control group; (**b**): Waterlogging stress for 12 days; (**c**): Waterlogging stress for 12 days and drainage for 6 days; (**d**): Waterlogging stress for 12 days and drainage for 12 days.

**Table 4.** Determination results of leaf epidermal stomata of *Phoebe sheareri* seedlings during the waterlogging-drainage process.

| Treatment | Stomatal Length (μm) | Stomatal Width (μm) | Stomatal Opening Rate (%) | Stomatal Density (n/mm²) |
|---|---|---|---|---|
| Control group | 13.35 ± 0.49 Bb | 7.35 ± 0.34 Aa | 27.55 ± 0.34 Aa | 2082 ± 22.47 Cc |
| Waterlogging stress for 12 days | 14.58 ± 0.33 ABb | 2.96 ± 0.20 Bb | 17.41 ± 0.76 Bb | 3298 ± 75.58 Aa |
| Waterlogging stress for 12 days and drainage for 6 days | 14.68 ± 0.53 ABb | 2.96 ± 0.14 Bb | 15.91 ± 0.36 BCbc | 3272 ± 53.80 Aa |
| Waterlogging stress for 12 days and drainage for 12 days | 16.16 ± 0.44 Aa | 3.04 ± 0.19 Bb | 15.22 ± 0.44 Cc | 2644 ± 70.95 Bb |

Compared with 12 days of waterlogging treatment, there was no significant change in leaf stomatal width after 6 days and 12 days of drainage. After drainage, the stomatal opening rate of seedlings decreased with the extension of drainage time. The stomatal opening rate of the *P. sheareri* seedlings drained for 12 days was extremely significantly lower than that of the control seedlings and of the seedlings under waterlogging stress for 12 days ($p < 0.01$). In terms of leaf stomatal density, there was an extremely significant difference between that of seedlings waterlogged for 12 days and that of seedlings drained for 12 days ($p < 0.01$), with a difference of 654 n/mm², but no significant differences between that of seedlings waterlogged for 12 days and that of seedlings drained for 6 days.

*3.3. Changes in Leaf Anatomical Structure of Phoebe sheareri Seedlings during Waterlogging Stress and Drainage Process*

3.3.1. Effects of Waterlogging Stress on Leaf Anatomical Structure of *Phoebe sheareri* Seedlings

With the extension of waterlogging stress time, the internal structure of *P. sheareri* leaves changed (Figure 4). The upper and lower epidermis cells of the control seedlings were arranged neatly and closely, the palisade tissue cells were plump and arranged neatly in a long column, and the spongy tissue cells were nearly round and arranged loosely (Figure 4a). There were no obvious changes in the anatomical structure of *P. sheareri* seedling leaves when they were waterlogged for 6 days (Figure 4b). After 12 days of waterlogging stress, palisade tissue cells of seedling leaves fused with unclear boundaries and loose arrangements, and the spongy tissue cells became smaller (Figure 4c). After 18 days of waterlogging stress, due to the narrowing of lower epidermis cells and the widening of upper epidermis cells, the palisade tissue cells shortened with increased intercellular spaces, and disorderly arranged spongy tissue cells (Figure 4d).

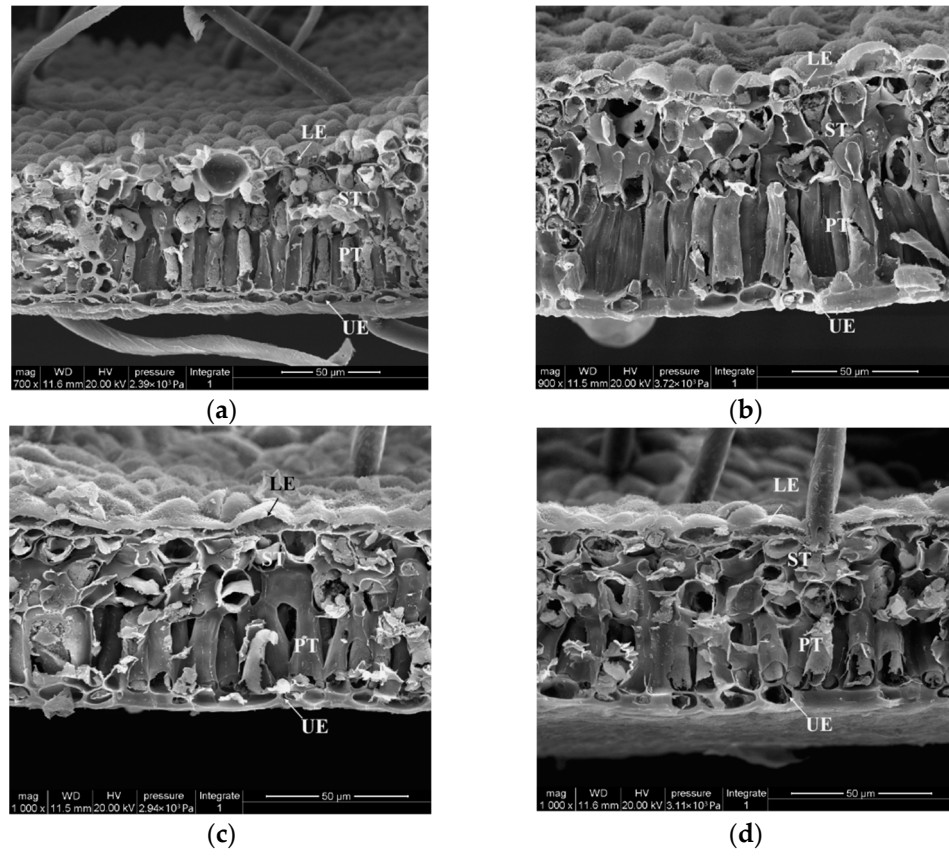

**Figure 4.** Changes in leaf anatomical structure of seedlings under waterlogging stress. (**a**): Control group; (**b**): Waterlogging stress for 6 days; (**c**): Waterlogging stress for 12 days; (**d**): Waterlogging stress for 18 days; LE: Lower epidermis; ST: Spongy tissue; PT: Palisade tissue; UE: Upper epidermis.

With the extension of waterlogging stress time, the total leaf thickness of *P. sheareri* seedlings witnessed an initial increase and then a decrease (Tables 5 and A3). The total leaf thickness of *P. sheareri* seedlings reached the maximum (88.58 μm) under waterlogging stress for 6 days, presenting a significant difference compared with those of seedlings at other waterlogging stress time nodes ($p < 0.01$).

**Table 5.** Determination results of leaf anatomical structure of seedlings under waterlogging stress.

| Treatment | Total Blade Thickness (μm) | The Thickness of the Upper Epidermis (μm) | The Thickness of the Lower Epidermis (μm) | Palisade Tissue Thickness (μm) | Spongy Tissue Thickness (μm) | The Ratio of Palisade Tissue to the Spongy Tissue |
|---|---|---|---|---|---|---|
| Control group | 77.15 ± 0.66 Bb | 5.94 ± 0.50 Aa | 9.10 ± 0.47 Aa | 28.49 ± 0.27 Bbc | 32.31 ± 1.01 BCb | 0.88 ± 0.02 Bb |
| Waterlogging stress for 6 days | 88.58 ± 1.40 Aa | 6.82 ± 0.76 Aa | 9.00 ± 0.17 Aa | 41.56 ± 0.52 Aa | 37.05 ± 0.99 Aa | 1.12 ± 0.03 Aa |
| Waterlogging stress for 12 days | 63.67 ± 0.43 Cc | 5.36 ± 0.18 Aa | 4.75 ± 0.70 Bb | 29.07 ± 0.41 Bb | 26.73 ± 0.87 Cd | 1.09 ± 0.05 Aa |
| Waterlogging stress for 18 days | 62.62 ± 0.75 Cc | 6.62 ± 0.48 Aa | 4.29 ± 0.67 Bb | 27.18 ± 0.78 Bc | 29.53 ± 0.47 Cc | 0.92 ± 0.02 Bb |

In the process of waterlogging stress, the upper epidermis thickness of *P. sheareri* seedling leaves did not change significantly. At 6 days of waterlogging stress, the thickness of the upper epidermis reached the maximum value (6.82 μm), with a difference of 0.88 μm compared with the control group. Compared to that of the control seedlings, the thickness of the lower epidermis of the seedlings' leaves under waterlogging stress showed a decreasing trend, during which the difference between the lower epidermal thickness of the seedlings waterlogged for 6 days and those of the seedlings enduring waterlogged stress for 12 days and 18 days reached a very extremely significant level ($p < 0.01$).

The leaf palisade tissue thickness of *P. sheareri* seedlings at the different waterlogging times was in the following order: waterlogging stress for 6 days > Control group > waterlogging stress for 12 days > waterlogging stress for 18 days. The palisade tissue thickness reached the maximum at 6 days after waterlogging treatment, and there was an extremely significant difference between other treatments ($p < 0.01$). The thicknesses of leaf spongy tissue among different treatments experienced significant differences ($p < 0.05$), and showed a trend of initial increase and subsequent decrease with the extension of stress time. Compared with the control group, the palisade tissue thickness increased by 4.74 μm after 6 days of waterlogging treatment, with extremely significant differences ($p < 0.01$).

The thickness ratio of palisade tissue to the spongy tissue of *P. sheareri* seedling leaves treated with waterlogging was larger than that of the control, and decreased gradually with the extension of waterlogging stress time. The ratio of palisade tissue to spongy tissue reached the maximum value at waterlogging stress for 6 days, and was extremely significantly higher than those at 18 days of waterlogging treatment and the control group ($p < 0.01$).

3.3.2. Effect of Waterlogging-Drainage Process on the Anatomical Structure of *Phoebe sheareri* Seedling Leaves

After 6 days of drainage, the cross-sections of upper epidermal cells on the leaf surface became wider and were nearly rectangular (Figure 5c). The cross-sections of lower epidermal cells become narrower with palisade tissues arranged loosely and the spongy tissue cells thickened (Figure 5c). After 12 days of drainage, the degree of damage of cells in each part of the seedling leaves was aggravated, the upper epidermis cells became narrower, and the palisade tissue cells became shorter, and, thus, the cell gap increased, the palisade tissue and spongy tissue cells fused with blurred boundaries, increased layer number and enlarged thickness of spongy tissue cells (Figure 5d).

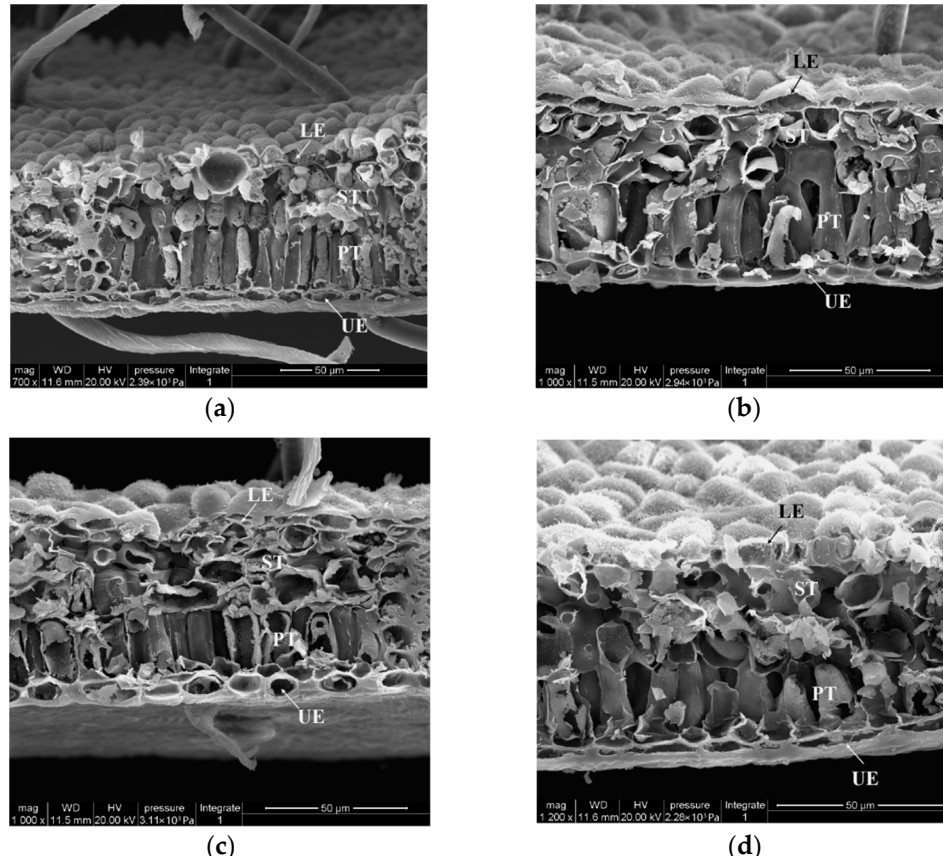

**Figure 5.** Changes in leaf anatomical structure of seedlings during the waterlogging-drainage process. (**a**): Control group; (**b**): Waterlogging stress for 12 days; (**c**): Waterlogging stress for 12 days and drainage for 6 days; (**d**): Waterlogging stress for 12 days and drainage for 12 days; LE: Lower epidermis; ST: Spongy tissue; PT: Palisade tissue; UE: Upper epidermis.

The total leaf thickness of the seedlings decreased gradually after drainage, and there was no significant difference between the seedlings after 12 days of waterlogging (Tables 6 and A4). The upper epidermal thickness of the leaves of *P. sheareri* seedlings showed a trend of increasing and then decreasing after drainage; in contrast, the lower epidermal thickness of the leaves showed a trend of decreasing and then increasing. The thickness of the upper epidermis reached its minimum value at 12 days of drainage, which was significantly different from other treatments ($p < 0.01$). The thickness of the lower epidermis reached the minimum value at 6 days of drainage, which decreased by 1.08 $\mu$m compared to 12 days of waterlogging stress and by 5.43 $\mu$m compared to the control. There was an extremely significant difference between the thicknesses of the lower epidermal layers of the leaves of *P. sheareri* seedlings drained for 6 days and those drained for 12 days ($p < 0.01$).

With the extension of drainage time, the thickness of leaf palisade tissue of *P. sheareri* seedlings decreased initially and then increased, and the thickness of palisade tissue of seedling leaves treated with drainage was extremely significantly smaller than those of the control seedlings and the 12-day-waterlogged seedlings ($p < 0.01$). The thickness of spongy tissue in the leaves of *P. sheareri* seedlings increased first and then decreased after drainage, which was extremely significantly different from that of the control seedlings ($p < 0.01$). Compared with waterlogging stress for 12 days, the spongy tissue thickness increased after 12 days and 18 days of drainage, by 3.12 $\mu$m and 2.47 $\mu$m, respectively.

**Table 6.** Determination results of leaf anatomical structure of seedlings during the waterlogging-drainage process.

| Treatment | Total Blade Thickness (μm) | The Thickness of the Upper Epidermis (μm) | The Thickness of the Lower Epidermis (μm) | Palisade Tissue Thickness (μm) | Spongy Tissue Thickness (μm) | The Ratio of Palisade Tissue to the Spongy Tissue |
|---|---|---|---|---|---|---|
| Control group | 77.15 ± 0.66 Aa | 5.94 ± 0.50 Aa | 9.10 ± 0.47 Aa | 28.49 ± 0.27 Aa | 32.31 ± 1.01 Aa | 0.88 ± 0.02 Bb |
| Waterlogging stress for 12 days | 63.67 ± 0.43 Bb | 5.36 ± 0.18 Aa | 4.75 ± 0.70 BCbc | 29.07 ± 0.41 Aa | 26.73 ± 0.87 Bc | 1.09 ± 0.05 Aa |
| Waterlogging stress for 12 days and drainage for 6 days | 63.16 ± 0.64 Bb | 6.08 ± 0.31 Aa | 3.67 ± 0.40 Cc | 22.92 ± 0.38 Bb | 29.85 ± 0.80 BCb | 0.77 ± 0.03 Bc |
| Waterlogging stress for 12 days and drainage for 12 days | 61.73 ± 0.74 Bb | 3.00 ± 0.14 Bb | 6.37 ± 0.73 Bb | 23.26 ± 0.69 Bb | 29.20 ± 0.80 BCbc | 0.80 ± 0.03 Bbc |

The thickness ratio of palisade tissue to spongy tissue in seedling leaves drained for 6 days was significantly different from that of control seedlings ($p < 0.05$), but the thickness ratio of palisade tissue to spongy tissue in leaves was not significantly different between the seedlings drained for 12 days and the control seedlings. The ratio of palisade tissue to spongy tissue decreased by 41.56% and 36.25%, respectively, after drainage for 12 days and 18 days, compared with waterlogging stress for 12 days.

### 3.4. Ultrastructural Changes in Mesophyll cells of Phoebe sheareri Seedlings during Waterlogging Stress and Drainage

3.4.1. Effect of the Ultrastructure of Leaf Mesophyll Cells of *Phoebe sheareri* Seedlings under Waterlogging Stress

With the extension of waterlogging stress time, the organelle structures, such as the structures of chloroplasts and mitochondria, in the leaves of *P. sheareri* seedlings changed to different degrees (Figure 6).

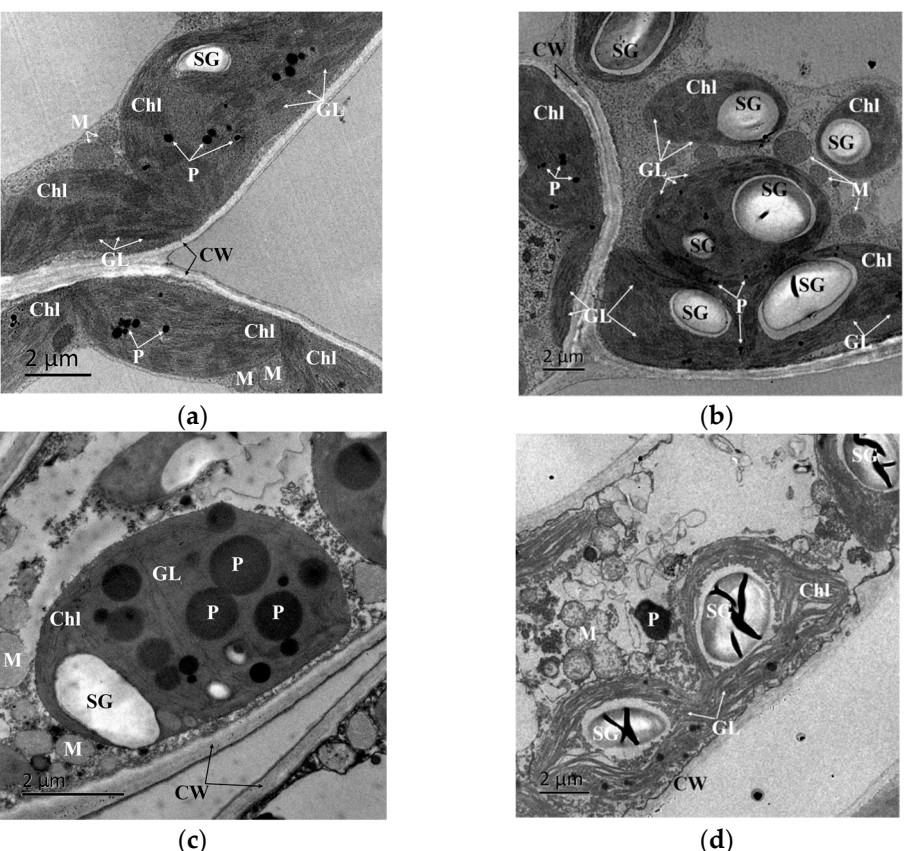

**Figure 6.** Ultrastructural changes of mesophyll cells of *Phoebe sheareri* seedlings under waterlogging stress. (**a**): Control group; (**b**): Waterlogging stress for 6 days; (**c**): Waterlogging stress for 12 days; (**d**): Waterlogging stress for 18 days; CW: Cell wall; Chl: Chloroplast; M: Mitochondrion; P: Plastoglobule; SG: Starch granule; GL: Granum lamella.

The cell walls of seedling leaves without waterlogged treatment were smooth, clear, and complete, and the cytoplasmic membrane was intact without folds (Figure 6a). The structure of the inner and outer membranes of the chloroplast was visible. It was fusiform and distributed in the inner membrane of the plasma membrane along its long axis with the thylakoids packed tightly, the grana lamellae arranged neatly, and a few plastid pellets and starch grains randomly distributed in the chloroplast. The mitochondrial structure was normal and complete, and its capsule visible, which was circular and distributed near the chloroplast with many inclusions.

After 6 days of waterlogging stress, the structure of the chloroplast membrane in seedling leaves was still clear and intact, but the chloroplasts were swollen in oval shapes, some chloroplasts had separated from the cell wall and moved toward the center of the cell, and the thylakoids were still stacked but not obviously so, the arrangement of the lamellae of some basal grains had changed, which might have been caused by the extrusion of starch grains (Figure 6b). The starch grains gradually became larger, some of them appeared with bright and dark stripes, and the mitochondrial structure was normal, distributed near chloroplasts, without obvious changes.

After 12 days of waterlogging stress, the chloroplast outer membrane of seedling leaf cells was no longer clear and there were impurities attached (Figure 6c). The chloroplast had clearly swelled, from spindle shapes to irregular ellipses. Plasmolysis appeared with blurred thylakoid structure and disorderly arranged grana lamellae. The plastid pellets had clearly swelled and increased. The outer membranes of the mitochondria were damaged, broken and incomplete, the contents of the mitochondria had spilled out, the matrix had become thinner, and the inner crest structure had vanished.

After 18 days of waterlogging stress, the chloroplast membrane of seedling leaf cells was wavy with rough edges, loose basal lamellae and partial dissolution (Figure 6d). Holes had appeared, and the starch grains in the chloroplast cells were irregular in shape, and distributed with many bright and dark stripes. Meanwhile, the plastid pellets were irregular and round. The mitochondrial outer membrane rupture was incomplete with impurities attached, some of the mitochondria merged, and some mitochondria had formed small vacuoles due to the break of the outer membrane and the outflow of the contents.

3.4.2. Effect of the Waterlogging-Drainage Process on the Ultrastructure of Leaf Mesophyll Cells of *Phoebe sheareri* Seedlings

The chloroplast outer membranes of leaf cells became rough with attached impurities when the seedlings were waterlogged for 12 days and drained for 6 days (Figure 7c). The chloroplasts swelled and distorted to an irregular oval shape. Meanwhile, their quantity decreased, and plasmolysis occurred. The thylakoid and grana inside the cells were blurred due to the disappearance of compact structure and the disorder of grana lamellae. The color of plastid pellets became lighter and dark patches appeared. The outer membranes of mitochondria were damaged, broken, and incomplete, and, thus, phenomena like the outflow of mitochondria inclusions, thinned matrix, and the disappearance of the inner crest structure occurred.

After 12 days of drainage, the leaf mesophyll cells were severely atrophied, and the outer membranes of the chloroplasts were wavy with impurities attached (Figure 7d). With intensified plasmolysis, the volume of plastid pellets became larger, the number and volume of starch grains increased, and some starch grains showed irregular shapes, due to the extrusion of plastid pellets. Impurities were attached to the outer membranes of mitochondria, and the structure of the inner crest disappeared.

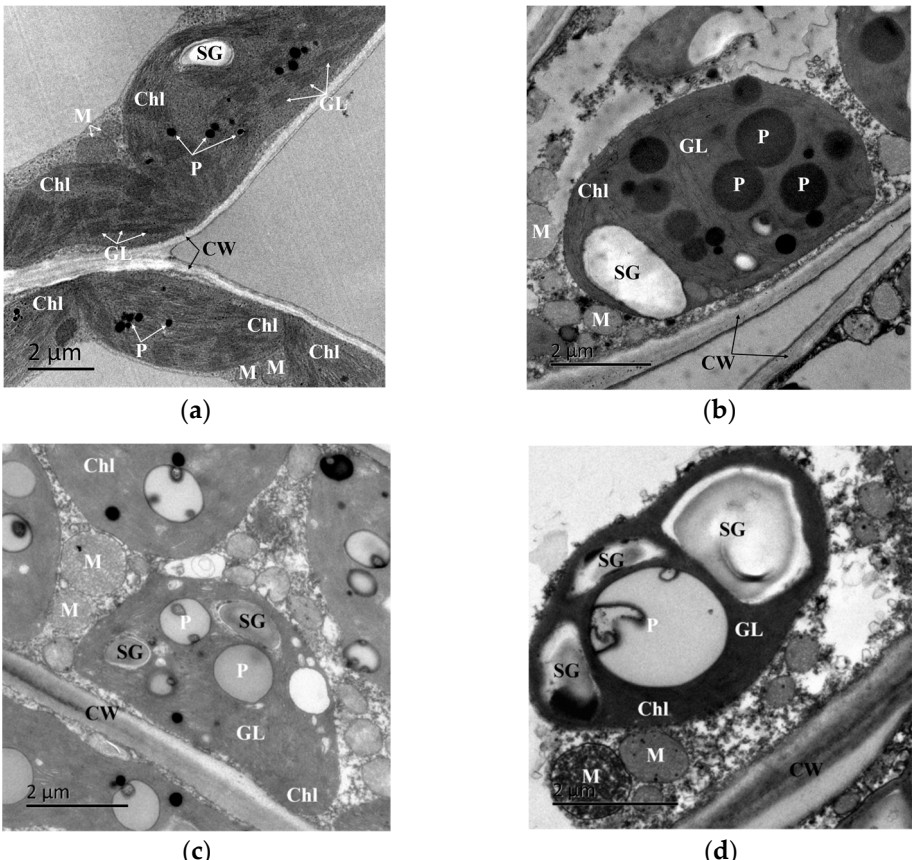

**Figure 7.** Ultrastructural changes of mesophyll cells of *Phoebe sheareri* seedlings during the waterlogging-drainage process. (**a**): Control group; (**b**): Waterlogging stress for 12 days; (**c**): Waterlogging stress for 12 days and drainage for 6 days; (**d**): Waterlogging stress for 12 days and drainage for 12 days; CW: Cell wall; Chl: Chloroplast; M: Mitochondrion; P: Plastoglobule; SG: Starch granule; GL: Granum lamella.

### 3.5. Correlation Analysis of Leaf Structure Indexes of Phoebe shareri Seedlings during Waterlogging Stress and Drainage

When the seedlings were drained after 12 days of waterlogging, the leaf anatomical structure of the seedlings witnessed significant change, which affected photosynthesis, respiration, and transpiration of leaves, as well as the growth and physiological metabolism of the seedlings. To investigate the relationship of leaf anatomical structure, correlation analysis was performed on related indicators of *P. shareri* seedling leaves during waterlogging stress and the drainage process, respectively (Tables 7 and 8).

When *P. shareri* seedlings were treated with drainage after waterlogging, there was a certain correlation between the stomatal structure and anatomical structure of leaves (Table 8). There was a significant negative correlation between leaf stomatal density and thinner epidermis thickness ($0.01 < p < 0.05$). The leaf stomatal opening rate was significantly positively correlated with the stomatal width ($0.01 < p < 0.05$). The total leaf thickness had an extremely significant positive correlation with the stomatal width and the stomatal opening ($p < 0.01$).

**Table 7.** Correlation analysis of leaf anatomical structure indexes of *Phoebe sheareri* seedlings under waterlogging stress.

|  | X1 | X2 | X3 | X4 | X5 | X6 | X7 | X8 | X9 | X10 |
|---|---|---|---|---|---|---|---|---|---|---|
| X1 | 1 | | | | | | | | | |
| X2 | −0.970 * | 1 | | | | | | | | |
| X3 | −0.945 | 0.973 * | 1 | | | | | | | |
| X4 | 0.907 | −0.981 * | −0.970 * | 1 | | | | | | |
| X5 | −0.560 | 0.744 | 0.740 | −0.856 | 1 | | | | | |
| X6 | 0.085 | 0.095 | −0.071 | −0.174 | 0.490 | 1 | | | | |
| X7 | −0.841 | 0.947 | 0.934 | −0.990 ** | 0.919 | 0.273 | 1 | | | |
| X8 | −0.063 | 0.298 | 0.337 | −0.477 | 0.855 | 0.540 | 0.589 | 1 | | |
| X9 | −0.344 | 0.557 | 0.584 | −0.707 | 0.966 * | 0.494 | 0.794 | 0.959 * | 1 | |
| X10 | 0.421 | −0.266 | −0.103 | 0.089 | 0.307 | −0.014 | 0.005 | 0.702 | 0.534 | 1 |

X1: Stomatal length (μm), X2: Stomatal width (μm), X3: Stomatal opening rate (%), X4: Stomatal density (n/mm$^2$), X5: Total Blade thickness (μm), X6: The thickness of the upper epidermis (μm), X7: The thickness of the lower epidermis (μm), X8: palisade tissue thickness (μm), X9: Spongy tissue thickness (μm), X10: The ratio of palisade tissue to the spongy tissue; ** indicates correlation at 0.01 level, * indicates correlation at 0.05 level, the same below.

**Table 8.** Correlation analysis of leaf anatomical structure indexes of *Phoebe sheareri* seedlings during the waterlogging–drainage process.

|  | X1 | X2 | X3 | X4 | X5 | X6 | X7 | X8 | X9 | X10 |
|---|---|---|---|---|---|---|---|---|---|---|
| X1 | 1 | | | | | | | | | |
| X2 | −0.767 | 1 | | | | | | | | |
| X3 | −0.846 | 0.985 * | 1 | | | | | | | |
| X4 | 0.336 | −0.862 | −0.779 | 1 | | | | | | |
| X5 | −0.843 | 0.991 ** | 0.997 ** | −0.790 | 1 | | | | | |
| X6 | −0.861 | 0.378 | 0.472 | 0.127 | 0.484 | 1 | | | | |
| X7 | −0.423 | 0.890 | 0.841 | −0.974 * | 0.835 | −0.077 | 1 | | | |
| X8 | −0.674 | 0.510 | 0.637 | −0.223 | 0.579 | 0.397 | 0.430 | 1 | | |
| X9 | −0.494 | 0.813 | 0.719 | −0.799 | 0.768 | 0.265 | 0.692 | −0.077 | 1 | |
| X10 | −0.226 | −0.099 | 0.055 | 0.315 | −0.019 | 0.158 | −0.095 | 0.804 | −0.654 | 1 |

** indicates correlation at 0.01 level, * indicates correlation at 0.05 level.

Under conditions of waterlogging stress, a certain correlation existed between leaf stomatal structure and leaf anatomical structure indices of *P. sheareri* seedlings (Table 7). There was an extremely significant negative correlation between stomatal density and the thickness of the lower epidermis ($p < 0.01$), the reason being that the stomatal density of *P. sheareri* concentrates in the lower epidermis, and change of stomatal density is closely related to change in lower epidermis thickness. The stomatal opening rate of the leaf epidermis was significantly positively correlated with stomatal width ($p < 0.05$), and significantly negatively correlated with stomatal density ($0.01 < p < 0.05$), indicating that the stomatal opening degree of *P. sheareri* seedlings was mainly affected by stomatal width and stomatal density during the waterlogging process. The thickness of spongy tissue was significantly positively correlated with the thickness of palisade tissue ($p < 0.05$).

## 4. Discussion

In the waterlogged environment, on the one hand, seedlings maintain their metabolic activity by consuming carbon reserves to facilitate the development of aerenchyma, adventitious roots, and hypertrophied lenticels, and, on the other hand, they decrease their metabolisms to preserve compounds for use in post-hypoxic conditions [29]. Waterlogging stress leads to hypoxia in the root system of seedlings, which, in turn, causes damage to the above-ground parts of seedlings, including leaves and other tissues and organs, through the accumulation of oxidation products [30,31].

### 4.1. Leaf Morphology

Under adverse circumstances, the change of leaf morphology is the most intuitive manifestation of the physiological state of seedlings. Changes in leaf shape can be environmentally induced, demonstrating that leaves are capable of responding to the surrounding climate conditions in a flexible manner [30]. Under conditions of waterlogging stress, the morphology of seedlings changes adaptively, the formation and growth of new leaves is blocked, the seedlings gradually age and fall off, and the growth of seedlings slows down [11]. Due to waterlogging stress, the leaves of *Pterocarya stenoptera* [32] and *Morus alba* [33] seedlings turned yellow, wilted, withered, and fell off. However, some studies have found that different plant species exhibit varying degrees of waterlogging symptoms due to their waterlogging tolerance. For example, after 18 days of waterlogging stress treatment, the lower leaves of *Populus* seedlings sensitive to waterlogging stress showed obvious chlorosis, while those tolerant to waterlogging stress had no significant change [34]. This paper demonstrates that, with the extension of waterlogging stress time, the leaves of *P. sheareri* seedlings gradually wilted, withered, and even fell off, which was similar to the research results of Kumutha et al. [35], Geng et al. [36], etc. In the early stage of waterlogging, the seedlings of *P. sheareri* grew new leaves and showed a short-term and recoverable wilting phenomenon. With the extension of waterlogging time, the wilting phenomenon of seedlings continued and could not be recovered. This was because the short-term waterlogging environment made *P. sheareri* seedlings mobilize their stress resistance mechanism to produce a large amount of energy and maintain the normal physiological metabolism of the seedlings, which was more conducive to the growth of *P. sheareri* seedlings. Waterlogging also had no significant effect on the growth and morphology of *Malus hupehensis* (Pamp.) Rehd. at the beginning of waterlogged treatment [37]. However, a long-term continuous waterlogging environment led to hypoxia in the roots of the seedlings, which made it impossible to synthesize and transform more energy to support growth, which was similar to the results of *Cerasus campanulata* after being waterlogged [38]. Consequently, the damaged metabolic mechanism in the seedlings blocked the synthesis of organic matter, inhibited the normal physiological metabolic process, affected the leaf morphology, and even led to leaf abscission [39].

This paper found that after waterlogging stress was relieved, the seedlings did not return to the state when waterlogged or before waterlogging. After 12 days of drainage, the seedlings appeared with the phenomenon of leaf abscission, and the survival rate declined steadily, which was different from *Taxodium* 'Zhongshansha 407', which survived without obvious symptoms of damage after relief of waterlogging [40]. The discrepancy results from *Taxodium* 'Zhongshansha 407' being a strong water-tolerant plant, maintaining a static watering-resistant strategy during the watering-out process [41,42]. However, it has also been shown that during the follow-up observation, grass plants, such as *Cynodon dactylon*, began to sprout new buds after the waterlogging environment was relieved, indicating that long-term waterlogging did not seriously affect their normal physiological functions, and the seedlings could recover growth through self-repair after the waterlogging stress was relieved [43]. The reason is that, although the leaf structures and functions of the seedlings were damaged under the condition of long-time continuous hypoxia, the overall growth of the seedlings was not affected, and survival could be guaranteed by adjusting the material and energy distribution of organs [40]. This suggests that the recovery process mainly depends on the type of species and nutrient reserves [44,45]. Plants with low levels of nutrient reserves cannot survive and grow with reserve substances under waterlogging for a long time [41].

### 4.2. Stomata and Anatomical Structure of Leaf Epidermis

The response of leaf epidermal structure to waterlogging stress is mainly reflected in the shape and arrangement of leaf epidermal cells, stomatal conductance, and stomatal density. The seedlings exchange gas, water, and signals with the external environment through the adjustment of stomatal size and density, which play an important role in the

regulation of stress resistance [46,47]. Leaves respond to stress by changing stomatal size, degrees of opening and closing, and density [48]. Leaf stomata are sensitive to the leaf water status. Stomata close actively to reduce water evaporation and increase waterlogging tolerance [12,49]. In this paper, it was found that waterlogging stress reduced the activity of guard cells in leaves of *P. sheareri* seedlings, and contributed to change of stomatal morphology and movement, increase in the ratio of length to width for stomata in the lower epidermis and stomatal density, as well as decreased stomatal opening degree, similar to the results of Gu et al. [50]. It is an adaptive strategy for plants, and under waterlogging conditions, plants growing in permanently damp environments have more stomata than those growing in dry habitats [49]. To maintain normal gas exchange and energy balance, the leaves quickly close part of the stomata, which may reduce leaf transpiration and the damage of waterlogging stress [51]. At the same time, the smaller the stomata, the higher the water use efficiency under conditions of favorable water supply [52]. The leaf stomatal status of *P. sheareri* seedlings gradually recovered after the waterlogging stress was relieved, but could not be restored to the state before waterlogging. The reason for this is that the waterlogging stress seriously damaged the stomatal state of the leaves and the internal cell structure, and even with drainage the stomatal status could not be recovered.

The leaf is the main organ for photosynthesis, respiration and transpiration of seedlings, and the unification of morphological structural change and function is the biological basis for seedlings to adapt to the growth environment [53,54]. It was found that under the condition of waterlogging, the upper epidermal cells, lower epidermal cells and palisade tissue cells of *P. sheareri* leaves transformed from a tight and neat arrangement to a loose arrangement. At the later stage of stress, the palisade tissue cells fused, the gaps became larger, and spongy tissue cells became smaller and were disorderly in arrangement, similar to the research results of Ren et al. [12]. Yin et al. [51] observed that intercellular spaces increased significantly in *Chrysanthemum zawadskii* leaves under waterlogged conditions, which facilitated rapid gas exchange. Intercellular spaces appeared in the leaf structure, which represented another positive adaptation to waterlogging, which could improve the capacity for oxygen capture, and help to store and exchange gases within the waterlogged tissues [55,56]. With the extension of stress time, the thickness of the upper epidermis and palisade tissue, and the ratio of the palisade tissue to spongy tissue of *P. sheareri* leaves experienced an initial increase and a subsequent decrease. This finding was different from that reported by Zhang et al. [16], who observed that the leaf thickness of *Sorghum bicolor* decreased under waterlogging stress, mainly due to changes in the thickness of the upper epidermal and mesophyll cells. The leaf thickness and its components increased differently after 6 days of waterlogging, which indicated that the stress-resistant mechanism of seedlings could respond quickly in a short time to maintain the normal growth of seedlings through leaf thickening, promoting the development of seedling leaves to a certain extent, similar to the results for *Chrysanthemum zawadskii* [49]. At the later stage of waterlogging stress, the leaf thickness was thinner due to decreased turgor pressure, which caused the enlargement of the cell gap and the loose arrangement of cells affecting the thickness of leaves [16].

After being relieved of waterlogging stress, the anatomical structure of *P. sheareri* seedling leaves confirmed that the degree of damage to all parts of the cells was aggravated, in that the upper epidermis cells became narrow, palisade tissue and spongy tissue cells were fused with blurred boundaries, the thickness of leaves and their components were not much different from those of the waterlogged seedlings, and even showed signs of gradual thinning. The reason is that although they grew in a normal water environment after drainage, the leaf structure of *P. sheareri* seedlings had been seriously and irreversibly damaged during the waterlogging process.

### 4.3. Mesophyll Cell Ultrastructure

Photosynthesis is directly influenced by the morphology and ultrastructure of chloroplasts, and in adverse circumstances, the obvious organelles in plant mesophyll cells are

chloroplasts and mitochondria [28]. Their morphologies and internal structures change with the external environment [12]. This paper found that under the conditions of waterlogging stress, the chloroplast membrane structure of seedling leaf cells ruptured gradually, the chloroplasts swelled continuously, deforming from spindles to irregular ellipses and separating from the cell wall with a tendency to move towards the center of the cell. The grana and thylakoids lost their compact and orderly arrangements and the grana lamellae gradually twisted. Thylakoid membrane degradation and lipid accumulation increased the number and volume of plastid pellets. Such changes also occurred in the waterlogging stress experiments of *Zea mays* and *Sorghum bicolor* [12,14]. The chloroplasts of *Kosteletzkya virginica* became spherical and their volume decreased under waterlogging. Moreover, the lamellae of thylakoids swelled, and chloroplast inclusions decreased [13]. Waterlogging damages the ultrastructure of mesophyll cells and reduces the photosynthetic capacity of the leaf [57]. The breakdown of membrane systems caused by waterlogging on mesophyll cells results in apoptosis and the loss of photosynthetic potential [58].

Compared with the seedlings waterlogged for 12 days, the ultrastructure of mesophyll cells in the seedlings drained for 6 days did not change significantly and did not show obvious signs of recovery. After 12 days of drainage, the chloroplast structure of leaf cells was further damaged, the outer membranes of chloroplasts blurred, the swelling of chloroplasts was aggravated, the separation of plasma wall was obvious, and a large amount of starch accumulated to form large starch granules, indicating that 12 days after the stress was relieved, the damage to chloroplasts was more serious than that caused by waterlogging. The reason is that the waterlogging environment caused irreversible damage to the chloroplast structure of *P. sheareri* seedlings. The chloroplast membrane gradually disintegrated, and grana and starch grains piled up together, resulting in the decline of the photosynthetic areas and hindering normal photosynthesis. The chloroplast structure could not be restored to the normal state after the stress was relieved [59]. In addition, this paper confirmed that there was a certain correlation between the abnormality of chloroplast structure and leaf color of *P. sheareri* seedlings during waterlogging progression. When waterlogged for 12 days, the young leaves of *P. sheareri* seedlings wilted, curled, and turned yellow. Among the chloroplast ultrastructures over the same period, it was observed that the thylakoid had a fuzzy structure, and the number and volume of plastid pellets increased, which was similar to the research results of Sun et al. [60] and Liu et al. [61].

As important organelles in respiration, mitochondria can not only provide energy for various life activities, but also participate in processes such as cell differentiation, cell information transmission, and cell apoptosis, with the ability to regulate cell growth and cell cycle [62]. Furthermore, certain mitochondria grow longer and become dysfunctional with time, and their membranes disintegrate [57]. After 12 days of waterlogging stress, the mitochondrial structure of *P. sheareri* seedling leaf cells began to change significantly, as the outer membranes broke and degraded, and contents flowed out. With the extension of stress time, the mitochondrial damage intensified, the membrane structure was damaged and degraded gradually, and the mitochondria fused. The phenomenon of mitochondria gradually swelling and finally disintegrating under waterlogging also occurred in *Kosteletzkya virginica* [11]. After the stress was relieved, the ultrastructure of mitochondria in leaf cells was not significantly improved, and the structure was still loose and fuzzy, indicating that the respiratory function of *P. sheareri* seedlings was damaged after drainage. Observation of *Cabernet sauvignon* cells found that vitamin C in mitochondria may flow out together with the contents, thereby causing programmed cell death (PCD), resulting in partial mitochondrial apoptosis and the formation of small vacuoles [63]. Some studies have also found that cell necrosis caused by programmed cell death of chloroplasts and mitochondria may be one of the mechanisms by which *Ammodendron argenteum* and *Triticum aestivum* seedlings adapt to water stress [64,65].

Under stress conditions, two organelles, the chloroplasts and the mitochondria, interact with each other. After the mitochondrial structure is destroyed, the metabolism of the seedling is disordered, unable to provide energy for the transport of photosynthetic

products and, thus, affecting the transport of photosynthetic products in the chloroplasts for seedling survival [66]. As a result, a large amount of starch accumulates in the chloroplasts, which cannot be transported in time. When the number of starch grains reaches a certain amount, it causes mechanical damage to the thylakoid structure, thereby affecting chloroplast photosynthetic rate and photosynthetic activity [57,67]. This paper confirmed that chloroplasts and mitochondria have different response times and degrees of response to stress. Chloroplasts began to respond in the early stage of stress in *P. sheareri* by changing their shape, size and membrane structure, and their systems became disordered and scattered, whereas mitochondria began to rupture the outer membrane, release contents and form small vacuoles in the late stage of stress. This asynchronous stress response alleviates the damage and ensures that *P. sheareri* seedlings can carry out physiological adjustments in a short time to maintain normal physiological metabolic functions. This pattern was also observed in *Populus simonii* under waterlogging stress, where chloroplasts were sensitive to stress, while mitochondrial structures were relatively stable [66]. This mechanism can only cope with stress for a short time and cannot play a role in a long-term continuous waterlogging environment. However, some studies have found that in the process of stress, mitochondria are less sensitive, and their stress response occurs later than that of chloroplasts since mitochondria need to provide senescent leaves with the energy required for various life activities, such as the transfer and transportation of nutrients [28].

Leaves are sensitive to environmental changes with strong plasticity, and the changes in their external morphology and anatomical structural characteristics can better reflect the adaptability of plants to waterlogging [17]. With the extension of waterlogging time, the balance between the organelles of *P. sheareri* seedlings was gradually broken, the chloroplasts were deformed, the plasma walls separated, the mitochondrial membranes were damaged, the inclusions were exuded, and the grainy lamellae were gradually distorted, resulting in leaf thickness changes [55,68]. The thickness of leaf components of *P. sheareri* gradually decreased under waterlogging conditions, especially the lower epidermis thickness. In addition, due to the influence of waterlogging environment on gas exchange, the stomatal opening of leaves also decreased gradually [48]. Under the influence of these factors, the leaf external morphology showed gradual wilting. When waterlogging stress causes irreversible damage to leaf cells, leaf stomata and leaf thickness gradually decrease, and organelles, such as chloroplasts and mitochondria, gradually disintegrate and cavitate, eventually causing leaf morphology to wither and seedling death [69,70].

Taken together, these results showed that leaf morphology, anatomical structure and ultrastructure associated with increasing waterlogging treatment respond differently. Nevertheless, leaf physiology or molecular mechanisms can further reveal the correlation between seedling leaves and changes in the external moisture environment. Additional measurements of leaf physiology, internal structure and moisture should be made in future studies to obtain further information.

## 5. Conclusions

This paper confirmed that waterlogging stress affects the external morphology of the leaves of *P. sheareri* seedlings, mainly reflected in aspects such as the inhibition of the growth of new leaves, gradual yellowing and wilting of leaves, and leaf abscission in severe cases. Further observation of the microstructure of the leaf epidermis illustrated that waterlogging stress leads to stomatal closure and increased stomatal density of the leaf epidermis, anda long-term waterlogging environment reduces the thickness of leaves and their components. Waterlogging stress mainly affects organelles such as chloroplasts and mitochondria, causing damage to chloroplast outer membranes, changes in shape, plasmolysis, disarray of grana lamellae arrangements, decrease of mitochondrial matrix concentrations, and disappearance of inner crest structures. After *P. sheareri* seedlings were treated with drainage, their leaf morphology, stomata, epidermal structure and mesophyll cells did not improve significantly, which was attributed to the fact that the seedlings suffered irreversible damage during the waterlogging process, and their internal metabolic

mechanism was seriously injured and could not be recovered. Based on the leaf morphology and various indices, the seedlings of *P. sheareri* performed best in several aspects such as leaf morphology, leaf epidermis microstructure and mesophyll cell ultrastructure when they were waterlogged for 6 days, which indicated that a short-term waterlogging treatment can promote better development of the leaves and their internal mechanisms in *P. sheareri* seedlings. However, under conditions of long-term waterlogging, the damage became increasingly serious.

This paper focused on preliminary observations of the changes in the external morphology of leaves, the microstructure of leaf epidermis, and the ultrastructure of mesophyll cells of *P. sheareri* seedlings under waterlogging stress and during the waterlogging–drainage process, but lacked deep exploration on the differences in the waterlogging resistance mechanisms among the internal organelles of *P. sheareri* seedlings and the dynamic changes in the internal metabonomics and molecular mechanisms of *P. sheareri* seedlings under waterlogging stress. In the future, it is still necessary to conduct in-depth research on the linkage mechanism of cell structure and function of seedling leaves in adverse environments, as well as on metabolomics and the molecular mechanisms.

**Author Contributions:** F.S. contributed to conceptualization, investigation, supervision, writing—review and editing. Z.P. contributed to conceptualization, data curation and visualization, writing—original draft, writing—review and editing. P.D. contributed to conceptualization, investigation, data curation and visualization, writing—review and editing. Y.S. contributed to supervision, conceptualization, review and editing. Y.L. and B.H. contributed to supervision, review and editing. All authors have read and agreed to the published version of the manuscript.

**Funding:** This research was funded by the subject of Key R&D Plan of Shandong Province (Major Scientific and Technological Innovation Project): Mining and Accurate Identification of Forest Tree Germplasm Resources (2021LZGC023); Jiangsu Agriculture Science and Technology Innovation Fund (CX (19) 2123).

**Data Availability Statement:** All data relevant to the study are included in the article.

**Conflicts of Interest:** The authors declare no conflict of interest.

### Appendix A

**Table A1.** ANOVA of leaf epidermal stomata of seedlings under waterlogging stress.

| Index | Square Sum | Degree of Freedom | Mean Square | F Value | *p* Value |
|---|---|---|---|---|---|
| Stomatal length | 5.647 | 3 | 1.882 | 1.617 | 0.225 |
| Stomatal width | 74.140 | 3 | 24.713 | 80.535 | 0.000 |
| Stomatal opening rate | 0.060 | 3 | 0.020 | 210.754 | 0.000 |
| Stomatal density | 6,649,454.502 | 3 | 2,216,484.834 | 106.044 | 0.000 |

**Table A2.** ANOVA of leaf epidermal stomata of seedlings during the waterlogging-drainage process.

| Index | Square Sum | Degree of Freedom | Mean Square | F Value | *p* Value |
|---|---|---|---|---|---|
| Stomatal length | 20.998 | 3 | 6.99 | 6.491 | 0.004 |
| Stomatal width | 67.621 | 3 | 22.540 | 72.143 | 0.000 |
| Stomatal opening rate | 0.052 | 3 | 0.017 | 312.298 | 0.000 |
| Stomatal density | 3,949,174.521 | 3 | 131,631.507 | 122.422 | 0.000 |

**Table A3.** ANOVA of leaf anatomical structure of seedlings under waterlogging stress.

| Index | Square Sum | Degree of Freedom | Mean Square | F Value | *p* Value |
|---|---|---|---|---|---|
| Total blade thickness | 2271.940 | 3 | 757.313 | 193.140 | 0.000 |
| The thickness of the upper epidermis | 6.616 | 3 | 2.205 | 1.618 | 0.225 |
| The thickness of the lower epidermis | 103.091 | 3 | 34.364 | 23.120 | 0.000 |
| Palisade tissue thickness | 674.194 | 3 | 224.731 | 163.064 | 0.000 |
| Spongy tissue thickness | 289.756 | 3 | 96.585 | 25.908 | 0.000 |
| The ratio of palisade tissue to the spongy tissue | 0.219 | 3 | 0.073 | 14.275 | 0.000 |

**Table A4.** ANOVA of leaf anatomical structure of seedlings during the waterlogging-drainage process.

| Index | Square Sum | Degree of Freedom | Mean Square | F Value | *p* Value |
|---|---|---|---|---|---|
| Total blade thickness | 802.069 | 3 | 267.356 | 93.849 | 0.000 |
| The thickness of the upper epidermis | 30.688 | 3 | 10.229 | 20.753 | 0.000 |
| The thickness of the lower epidermis | 83.573 | 3 | 27.858 | 15.861 | 0.000 |
| Palisade tissue thickness | 146.466 | 3 | 48.822 | 29.286 | 0.000 |
| Spongy tissue thickness | 78.828 | 3 | 26.276 | 6.895 | 0.003 |
| The ratio of palisade tissue to the spongy tissue | 0.301 | 3 | 0.100 | 15.970 | 0.000 |

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
