# Peer review of "Effect of Waterlogging Stress on Leaf Anatomical Structure and Ultrastructure of Phoebe sheareri Seedlings"

_forests, doi:10.3390/f14071294_

Round 1
Reviewer 1 Report
The manuscript titled "Effect of waterlogging stress on leaf anatomical structure and ultrastructure of Phoebe sheareri seedlings" investigates the effect of waterlogging on external leaf morphology, leaf epidermis microstructure, and mesophyll cell ultrastructure in Phoebe sheareri seedlings. 2-year-old seedlings were exposed to different durations of waterlogging stress (6, 12, 18 days), and water was drained for 6 or 12 days only for 12-day waterlogging treatment seedlings.
As the authors noted the high value of the species in China, this paper might offer some insights into how the species respond to waterlogging that frequently occurs/ will occur around the Yangtze River region. However, despite the new insight into the species' response to waterlogging focusing on their leaf morphology, leaf epidermis microstructure, and mesophyll cell ultrastructure, the paper has too many structural issues, which makes it hard to follow the argument. For instance, the description of methodologies such as experimental design and analysis (how ANOVA assumptions were tested etc.), and treatment were insufficient to repeat the experiment. Moreover, the authors are confused with the content of the result section and discussion section. The language sentence structures of this manuscript are incomprehensible. The paper needs rewriting and thorough language editing for a proper peer review.
Here are some points to be improved:
1. Introduction
In the introduction section, it would greatly benefit the readers if the authors could provide more detailed information regarding the past studies cited. Specifically, it is crucial to elucidate the tested species and how they responded to the treatment. This additional information will enhance the understanding of the existing knowledge in the field and set the stage for the current study.
I recommend the authors consider the following points when discussing the past studies in the introduction:
- Clearly mention the species or organisms that were the subject of the previous studies, including their scientific names.
- Describe the experimental procedures and treatments applied to those species.
- Highlight the key findings and outcomes of those studies in relation to the research topic of the current paper.
- Discuss any knowledge gaps or limitations identified in the previous research that the current study aims to address.
By incorporating these details, the introduction will provide a comprehensive overview of the relevant literature and establish a strong rationale for the present study. It will enable readers to grasp the research's significance and appreciate its contribution to the existing knowledge base.
2. Materials and Methods
The Materials and Methods section lacks crucial information regarding the experimental design. To properly assess the relevance and validity of the study, it is essential to provide a comprehensive description of the experimental design employed. This includes the following key elements:
- Clearly state the overall design framework to establish the study's nature and approach.
- Specify the specific variables or factors investigated and their respective levels or categories.
- Explain how the study subjects or samples were selected or assigned to different treatments or groups.
- Provide details on the replication and sample size, including the number of independent replicates or observations for each treatment or group (e.g., If 30 seedlings were exposed to 12-day waterlogging treatment and then used for another draining experiment, how many seedlings were tested for the draining experiment?)
By addressing these points, readers will clearly understand the experimental design and be able to evaluate the robustness and reliability of the study's findings. I recommend revising the Materials and Methods section to include these important details, as they are essential for the reproducibility and interpretation of the research."
3. Results
4. Discussion
There appears to be some confusion between the Results and Discussion sections of the manuscript. The Results section should present the study's findings clearly and concisely, providing a straightforward description of the observed data without interpretation or speculation. On the other hand, the Discussion section should interpret the results, compare them with relevant literature, and provide a deeper analysis and context for the findings.
To improve the clarity and structure of your manuscript, I suggest the following:
- Ensure that the Results section focuses solely on presenting the observed data without any interpretation or speculation about their meaning.
- Organize the results logically, using appropriate tables, figures, and statistical analyses (some captions are missing and mislocated).
- Reserve the Discussion section for interpreting the results and placing them in the context of the existing literature. Discuss the implications, limitations, and potential mechanisms underlying the findings.
- Address any discrepancies or similarities with previous studies and comprehensively analyze the results.
Ensure there is a clear transition between the Results and Discussion sections, indicating where the presentation of results ends, and the interpretation and analysis begin.
By separating the Results and Discussion sections and adhering to their respective purposes, readers will be able to follow the progression of the study more effectively and appreciate the significance of the findings in the broader context.
I recommend reviewing the manuscript by a professional native speaker to ensure language accuracy and clarity. While the scientific content of the study appears valuable, there are some areas where the language and grammar could be improved for better readability and understanding.
A native speaker with expertise in scientific writing can help refine the manuscript, ensuring that the language is precise, the terminology is used correctly, and the overall structure and flow of the document are enhanced. This review process can greatly contribute to the overall quality and professionalism of the manuscript.
Reviewer 2 Report
The specific modification suggestions are as follows:
1. In the introduction section, the significance and purpose of the research are not very clear, so it needs to be further elaborated.
2. The study adopts the "double pot method" to artificially simulate waterlogged environment, and needs to explain the "double pot method" through text, pictures, and other means.
3. The indexes of correlation analysis mainly include stomata and leaf thickness in the leaf anatomical structure. It is suggested to modify the title and content into correlation analysis of leaf anatomical structure indexes of P. sheareri seedling.
4. In the discussion section, the changes of leaf morphology, anatomical structure and ultrastructure of P. sheareri seedlings under waterlogging treatment were combined to systematically describe the impact of waterlogging stress on the overall leaf structure of P. sheareri seedlings
Minor editing of English language required
Round 2
Reviewer 1 Report
Please see the comment below:
Abstract
Line 13: Little~ : Please specify “leaf sites”.
Introduction
· Please provide a hypothesis of your study in the introduction.
· Although there are many studies done on tree species investigating responses to waterlogging, the authors provided literature that tested on non-tree species such as Z. mays and Sorghum bicolor. Please provide more literature that focuses on tree responses.
Line 30: Please provide the name of the city and country after “Yangtze River”.
Line 52: Change “The pots of Sorghum bicolor were placed in another pot with no holes at the bottom to be waterlogged for two weeks, the leaf thickness of S. bicolor was significantly decreased, which was mainly due to changes in the thickness of the upper epidermal and mesophyll cells [16]” to “In the previous study, after seedlings of Sorghum bicolor were waterlogged for two weeks, leaf thickness was significantly decreased, mainly due to changes in the thickness of the upper epidermal and mesophyll cells [16]”
Materials and Methods
Line 133:
· Table 1 seems redundant to line 116 ~" in your manuscript
· Line 108: The content of 2.2. Experimental design seems to be “Waterlogging treatments” rather than “Experimental design”. I did not find any experimental design in this manuscript. Please provide the experimental design you used for this experiment.
· What does “area” mean in line 121, 122, and 123?
· Change “plants” to “seedlings” in line 122 and 123.
· What CK stands for? Control K…..?
· Line 135~ should be in the next section, “Indicator determination”.
Line 146:
· Provide the detail of your research methodology. I see the authors cited Jain et al. [24]. However, there was no further detail. Citing a paper without additional information can confuse readers and hinder the study's reproducibility. Including a brief summary or description of the method will help readers understand the approach and ensure transparency in the research process. For instance, how were the samples prepared for SEM? Also, when describing a scanning electron microscope (SEM) in a research paper, it is essential to provide relevant details about the specific SEM instrument used. When these parameters were measured, how often? This information helps to establish the credibility and reproducibility of the study. This comment goes to line 153 also.
Line 148: Provide more details for the number of replications for your measurement. For instance, how many leaves were taken from one seedling?
Line 151: Do not use “etc”. Write them all.
Line 156:
· The authors stated that the data statistics were performed with Excel 2018, but ANOVA was analyzed using SPSS. I am not sure what was analyzed using Excel 2018.
· As I mentioned in the first review comment, there was no description of how the data was tested for ANOVA assumptions. Did you check the ANOVA assumption before employing the data analysis?
Results
· I assume the authors are still confused with the contents of the result and discussion section. For instance, in Line 177~ (not only here, but there are more similar confusions made in the result section), the authors explain the reasons for the result, which should be discussed in the discussion section. Based on my previous comments, Please ensure what should be in the result section and discussion.
· Be more concise in your presentation of results. Focus on providing essential information without unnecessary repetition or excessive detail. Please remove any repetitive phrases or redundant statements to streamline the content.
· (Tables 3, 4, 5, and 6) There are uppercase and lowercase for significant differences, but I do not understand why. As my understanding, the effects of the treatments were tested on all data using the one-way ANOVA with Duncan's multiple range tests at p < 0.05, correct? And the letters are from Duncan's multiple-range tests, correct? In that case, I am unsure why the tables have uppercase and lowercase.
· Provide an ANOVA table for each analysis that contains F and P values and degree of freedom.
· The authors use “It can be seen from” in the result section several times. This can be deleted and add the Figure number at the end of the statement.
· I recommend the authors combine the explanation of pictures and the result of the analysis in the same paragraph. For instance, Line 259~ can be mentioned in Line 271~298. Ensure to adapt this advice to the entire result section.
· Change all of “0.01<P<0.05” to “P<0.05”.
· Delete all (P>0.05). Readers would know when you state that there was no significant difference between treatments, the P value is P> 0.05, this is standard. Plus, you mentioned that in 2.4. Statistical analysis already.
Line 165: Since the data were not statistically analyzed, better to avoid using the word “significantly”.
Line 182:
· (Table 2) Provide horizontal lines between each experimental treatment. It is hard to read, especially the leaf morphology part.
· (Table 2: Leaf morphology) In order to enhance the precision and scientific rigor of the manuscript, I would recommend replacing qualitative expressions such as 'seriously,' 'more,' and 'slightly' with quantitative expressions or specific numerical values. This will provide readers with more precise and measurable information. For example, instead of using 'more,' authors could specify the exact percentage change observed. Quantitative expressions will make the findings more precise and facilitate better interpretation and comparison.
Line 187~: Not only here but also the throughout the result section, provide the alphabet of the figure you are talking about in the sentence, for example, “(Figure 2a)”. Providing an alphabet for each made it easier for us to understand. Ensure to adapt this advice to the entire result section.
Line 209: Again, this statement is not for the result section, and if you state this in the discussion section, you need literature to support the statement.
Line 210~:
· Rewrite the paragraph. You have to state what you found from the analysis (Table 3). This is the confusing part because of the two letters, upper and lowercase, and the lack of an ANOVA table, and the authors are not reporting the results properly throughout the result section. Ensure to provide an ANOVA table and what the letters mean in the tables.
Line 229: Delete “It can be seen from the table” and add the figure number at the statement's end.
Line 229: Rewrite the paragraph. It is hard to read. The line 233-244 as well.
Line 235:
· What is natural drainage? Was there a treatment called “natural drainage,” ?
· What do you mean by gradually? How frequently measured the stomatal opening rate? Provide it in the methodology section. Line 239 as well.
Line 235~: Most of the report here mentions significant differences between treatments but not specific quantitative information about the observed differences. For example, they can mention how much lower or higher one treatment group is compared to another and provide the mean differences between groups to quantify the magnitude of the differences. This can be applied throughout the result section. Ensure to adapt this advice to the entire result section.
Line 250~255: Delete the paragraph. I suggest the authors focus on the effect of waterlogging treatments.
Line 260: Delete “obviously”.
Line 263: Indicate which picture you are referring to at the end of the statement, for instance, “~arranged loosely (Figure 4a). Ensure to adapt this advice to the entire result section.
Line 298: Unsure what is 1.12.
Line 299: Phoebe sheareri should be italic.
The same as the previous comment
Author Response
Thank you for your suggestions. Please see the attachment of the reply to your comments.
